# "They never appear on TV and if they have, I might have missed that moment." How publics in South Africa and Germany view visible scientists

Simone Rödder[1]*, Lars Guenther[2,3], Marina Joubert[3]

1 Department of Social Sciences, University of Hamburg, Hamburg, Germany, 2 Department of Media and Communication, LMU Munich, Munich, Germany, 3 Centre for Research on Evaluation, Science and Technology (CREST), Stellenbosch University, Stellenbosch, South Africa

* simone.roedder@uni-hamburg.de

## Abstract

Academic interest in scientists who regularly appear in the media dates back to Rae Goodell's seminal book "The visible scientists", in which she lists distinct characteristics of visible scientists, including being controversial, articulate, colorful, and reputable as a scientist. Visible scientists thus share relevant media-related characteristics that stand out in their portrayal as a group and are reminiscent of other celebrities' characteristics. However, questions arise: what is special about the celebrity being a scientist? How many and what types of scientists fall into this category? What are the peers' and the public's expectations towards the social role of the visible scientist? To date, work on visible scientists has focused on theorizing them in the context of the relationship between science and its publics and empirical studies have mainly sought to characterize visible scientists and focused on single countries. This paper fills research gaps on the public perception of and expectations towards visible scientists as well as comparative studies by surveying publics in Germany and South Africa. Our data shows that Goodell's criteria also apply to how respondents see and expect visible scientists to be. Interestingly though, a majority of non-responses to the request to name up to three visible scientists currently living in the respective country demonstrates that, overall, scientists are rather invisible to the public. Visible scientists remain a rare phenomenon despite changing media environments and a recent pandemic. In conclusion, we suggest that "visibility" (rather than "celebrity") is the more appropriate term to refer to the temporary phenomenon of scientists who become visible in the public sphere.

## 1 Introduction

On June 3, 1881, the French correspondent of the *London Times* described the results of a public experiment with which Louis Pasteur established himself as one of the leading scientists of his time. To demonstrate the principle of immunity through vaccination, Pasteur infected

**Data Availability Statement:** Mede, N. G., Cologna, V., Berger, S., Besley, J., Brick, C., Joubert, M., Maibach, E. W., Mihelj, S., Oreskes, N., Schäfer, M. S., van der Linden, S., Abdul Aziz,

N. I., Abdulsalam, S., Abu Shamsi, N., Aczel, B., Adinugroho, I., Alabrese, E., Aldoh, A., Alfano, M., . . . Zwaan, R. A. (2024). Perceptions of science, science communication, and climate change attitudes in 67 countries: the TISP dataset. Preprint. https://doi.org/10.31234/osf.io/jktsy.

**Funding:** The author(s) received no specific funding for this work.

**Competing interests:** The authors have declared that no competing interests exist.

sheep, goats, and cows with the anthrax bacillus on a farm in Pouilly-le-Fort, France (quoted in [1]). The historical perspective demonstrates that addressing non-scientific audiences with evidence during the research process is not a new phenomenon; visible scientists *per se* are, therefore, not a product of modern media cultures. However, anyone experimenting with a microbe on their farm only becomes visible to a wider public when news media reports on the proceedings or by successful self-mediation. Today's publicly visible scientists can thus be described as a phenomenon of relevance in and across multiple digital and non-digital media environments. By ´public visibility´, in this paper, we mean scientists who regularly appear in the media, including on television, print and online news, radio, and social media platforms. We refer to these scientists as *visible scientists*.

Academic interest in visible scientists dates back to Rae Goodell's seminal book "The visible scientists", in which she characterizes the visible scientist as "relevant, controversial, articulate, colorful, and reputable as a scientist" [2]. Fahy [3] describes what he calls celebrity scientists as stars that grip the public imagination. He adds that what matters is how they communicate, how their science is tied to public issues, how interesting they are as personalities but also, a compelling presence and good looks. The visible scientists thus share some important media-related characteristics that stand out in their portrayal as a group and are reminiscent of other celebrities' characteristics. For science communication scholars, relevant questions arise about the particularities and circumstances of the fact that the celebrity is a scientist: How many and what kinds of scientists fall into that category? Beyond personal and individual attributes, what are essential characteristics of the social role of a visible scientist? And how are these perceived by different publics?

To date, work on visible scientists has focused on theorizing them in the context of the relationship between science and its publics [4] and empirical work has mainly sought to characterize visible scientists [3, 5, 6] and focused on single countries (but see [7, 8] for exceptions). However, the public perception of visible scientists and the public's expectations towards this social role have not been studied yet. In this paper, we aim to fill this research gap by exploring the public perception of visible scientists in a cross-country comparative design. We chose Germany as a country from the Global North and South Africa as a country from the Global South for this study.

The paper is structured as follows: We start off by further introducing the phenomenon of visible scientist, review the literature, derive research questions, and motivate the case approach and country selection (2). We then introduce the methodology and data corpus (3), and present our results on the public perceptions (4.1) as well as public expectations towards visible scientists (4.2). We contextualize our findings with the literature, spell out some avenues for further research (5), and end with a conclusion on the appropriateness of terms (6).

## 2 The social role of visible scientist

### 2.1 Theorizing and characterizing visible scientists

Conceptual work has theorized the visible scientist as a social role, namely as one among several in the role-set of scientists [4]. The expectation that scientists should be visible has been omnipresent for many years ("Thou shalt communicate"–[9]) and public engagement (or third-mission duties) are now among many scientists' activities (e.g., [10]). However, it is up for discussion whether repeated and longstanding public visibility has become a widespread phenomenon or remains restricted to a few emblematic cases, including German physicist Albert Einstein, US-American astronomer Carl Sagan, and South African heart surgeon Christiaan Barnard. Working scientists have asked whether this kind of celebrity is "bad for science or good for society?" [11], arguing that "celebrity is an opportunity that should not be

squandered. Scientists who become recognizable have a chance and perhaps even a responsibility, which they have often exploited, to promote science literacy, combat scientific nonsense, motivate young people, and steer public policy discussions toward sound decision making." Recently, the science communication demands of the COVID-19 pandemic have produced public faces or "Pandemic-ons" [12, 13] in most countries, including virologist Christian Drosten in Germany and HIV expert Salim Abdool Karim in South Africa.

Methodologically, Goodell looked at "the 'fittest' in the species" [2], including Einstein and Sagan, to conclude that visible scientists have distinct characteristics, and that they have more in common with other celebrities than with the average scientist, namely their positive media orientation. A particularity distinguishing them from other celebrities, is that they are typically reputable scientists, albeit a recent study finds that visibility during the COVID-19 pandemic corresponded with low reputation in Italian media [13]. Academic reputation is therefore one, but only one of several conditions for becoming visible in the media (see also [14, 15]). Visible scientists have been shown to link science and its publics by circulating new ideas, sparking social movements, and shaping public and policy debates [3, 5, 6]. Also, a certain willingness to work with the media has been noticed from early on. Visible scientists focus attention related to a scientific topic, problem, or research field, on their person, and it is precisely in controversies and conflict situations that they seek visibility [16].

For the news media, media prominence itself becomes newsworthy, as evidenced by the fact that some people are 'just' celebrities: they are famous for being famous. An example is US-American media personality Kim Kardashian, whose media presence includes hundreds of millions of followers on X (formerly Twitter) and Instagram. In contrast, there are celebrities who belong to the elites of relevant social worlds, such as powerful politicians, the best athletes, or the Pope, and subsequently become prominent, i.e., their political, religious, or economic role is their nomination to prominence [17]. In science, the Nobel laureates, publicly acclaimed each year in November, are a case in point [18].

A key insight from sociological research is the "ambivalence of visible scientists" [4], i.e., the tension between a scientist's public visibility and the expectations of peer communities about what counts as valuable science, who should control academic capital and decide on academic reputation. The relevance of reputation within the reward system in science socializes scientists to care about visibility in their peer community, i.e., recognition among those with similar academic specializations. This implies that peers have control over the work of their colleagues through awarding them recognition, which is arguably also consequential for their media contacts. This is illustrated by the following quote, taken from Russo [19]: "You want to communicate this stuff," she says, "but always there's this little scientist on your shoulder saying 'You can't say that'." Underlying these considerations is thus a perspective that theorizes science and journalism as professional fields and that foregrounds the analytic distinction between academic reputation on the one hand and media prominence on the other. Theorizing academic reputation and media prominence as distinct forms of social capital in science and the media, respectively, is interesting because they compete as soon as a scientist becomes publicly visible, raising empirical as well as normative questions about interactions, blurring, or conversions. For this reason, it is of sociological interest to study both perceptions of as well as expectations towards visible scientists across peer and public audiences.

In sum, conceptual and empirical work on visible scientists across centuries and countries has elucidated many aspects and similarities that characterize the social role of the visible scientist. However, the public perception of visible scientists as well as the public's expectations towards this social role have not been studied yet. To address this research gap, we ask RQ1: *What are the perceptions and expectations that publics have of publicly visible scientists in Germany and South Africa?*

## 2.2 Linking trust in science and media use to public perceptions

Focusing on the public perception of visible scientists, as well as the public's expectations towards this social role, allows for a comparison of related concepts, of which, in this study, we make connections to *trust in science* and *science-related media use*. For some time now, a supposed decline of public trust in science and its implications have been discussed–often linked to digital media environments [20, 21]. This makes the connection to perceptions of and expectations towards visible scientists all the more interesting. Public trust in science, and especially epistemic trust (see also [22–25], plays a crucial role in the relationship between science and its publics. In an online survey of American publics, the warmth and competence perception of different professions were studied [26]. The results show that job holders such as scientists, researchers, engineers, and professors were ambivalently perceived as high in competence and low in warmth, i.e., they earn respect but not necessarily trust, and respondents report mixed emotions that include both admiration and resentment. The authors conclude that "being seen as competent but cold might not seem problematic until one recalls that communicator credibility requires not just status and expertise (competence) but also trustworthiness (warmth)" [26]. Other than in this study, trust in science is often defined as a multilevel [27, 28] and multidimensional construct [20, 22–24]. Multilevel means that there is a distinction between science as a social world (i.e., macro-level), scientific organizations (i.e., meso-level), and scientists (i.e., micro-level; see also [29]. Assessments of visible scientists clearly link to the micro-level. Multidimensional means that when referring to epistemic trust in science, there are several established dimensions underlying this construct, such as expertise, integrity, benevolence, transparency, and dialogue [20, 23, 24] (see also [22])–all of them need to be considered in assessments of what makes scientists trustworthy and when establishing a connection to perceptions of and expectations for scientists' public visibility.

At the same time, audiences' assessments of scientists' visibility are probably related to their media use. Only through (mediated) exposure to scientists can perceptions and expectations of visible scientists take shape. Notably, in many countries, including Germany and South Africa [7], large parts of the public who traditionally used journalistic media for information about science, are now obtaining this information online, increasing the societal relevance of social and fringe/populist sources. Hence, in digital media environments, among other options, people can encounter visible scientists through journalistic media but also follow them directly on social media. We consider all of this and ask RQ2: *What is the link between perceptions and expectations of visible scientists, trust in scientists, and media use?*

## 2.3 Studying perceptions and expectations in Germany and South Africa

Many previous studies focus on single-country cases. We address this gap in comparative research by introducing a cross-country comparison and surveying publics in two countries. We chose Germany and South Africa as cases, one belonging to the Global North and one belonging to the Global South. Besides fundamental geopolitical and socioeconomic differences, these two countries share some similarities that are relevant for this study. Both have democratic political orders with free media and social media, which publish predominantly in German and English respectively. Both have established science and higher education systems and active collaborations between them, as well as academic ties with many other countries. Eventually, both have cases of iconic visible scientists. In Germany, physicist Albert Einstein won the Nobel prize in 1921 for his insights into relativity theory, turning his well-known counterfeit into a synonym for a scientist, as well as science: "Einstein is relativity" [30]. In South Africa, the surgeon Christiaan Barnard successfully transplanted the first human heart in a Cape Town hospital in 1967. He remained a skilled and prolific surgeon for many years,

but also became an international celebrity and a public persona as South Africa's "fallible king of hearts" [31, 32]. During the recent COVID-19 pandemic, Germany and South Africa also had high-profile scientists: In Germany, virologist Christian Drosten had a regular podcast, "Coronavirus Update", in which he informed the public about pandemic-related issues, and HIV/Aids expert Salim Abdool Karim became a household name in South Africa for his high media profile during the pandemic [12]. We therefore analyze RQ1 and RQ2 with a focus on *similarities and differences between German and South African publics' perceptions of and expectations towards visible scientists.*

## 3 Methodology and data

### 3.1 Design and sampling procedure

This study is part of the crowd-sourced Many Labs project "Trust in Science and Science-Related Populism" (TISP, for a project overview, see [33], for the data set [34]). Institutional Review Board approval was registered at Harvard University. Here, we only include the surveys conducted with samples in Germany and South Africa, for which we added additional questions about perceptions and expectations of scientists' visibility to the standard questions and items, which were mainly on trust in science and scientists.

To answer the research questions, we deduced ten features of publicly visible scientists from the literature reviewed above, as well as from our previous work [11]. These include the most established common characteristics of visible scientists, namely that they *are senior, are male, have a good scientific reputation, are charismatic (likeable and inspiring leaders), are media-savvy (work well with the media), are highly articulate (well-spoken, use accessible language), share details about their private lives with the public, are controversial, and are criticized for seeking the limelight.* We were also interested in the extent to which scientists with high public profiles match the expectations that the public might have about them. We therefore asked for the expectations regarding the above characteristics, i.e., visible scientists *should be senior, male, controversial* etc. The complementary open-ended and closed-style questions in the survey cover important theoretical aspects but also allow people to express their views and concerns in their own words.

The data was collected via online surveys in German (for German respondents) and English (for South African respondents) throughout February 2023. The TISP project provided the English survey, and the translation into German was done by native speakers who were involved in the research. The surveys were programmed via Qualtrics survey software and made use of online panels of the market research company Bilendi & Respondi. Participants were incentivized by this company and provided written informed consent for their participation. Additional information regarding the ethical, cultural, and scientific considerations specific to inclusivity in global research is included in the S1 File.

Initially, we collected 1.551 responses in Germany and 2.225 in South Africa; data cleaning (e.g., completing the survey, passing two attention checks) left 1.011 responses for the German case, and 1.027 responses for the South African one. Since quota samples were used (five bins for age: 18–29 years, 30–39 years, 40–49 years, 50–59 years, 60 years and older, 20% each; two bins for gender: male, female, 50% each), data were weighted according to national distributions of age, gender, and education level; since this data was not always provided, this process resulted in a final sample size of 1.000 responses for each country; hence, 2.000 responses in total.

### 3.2 Survey protocol

The survey, each in German and English, was administered as proposed in the pre-tested TISP project (i.e., items on trust in scientists), while the authors were able to add additional

questions (i.e., perceptions and expectations of scientists' visibility). In the following, we focus on aspects of the survey protocol used in the present study. The survey started with participants reading and conforming to a consent form and providing sociodemographic information. Next, definitions of science ("the understanding we have about the world from observation and testing") and scientists ("people who study nature, medicine, physics, economics, history, and psychology, among other things") were provided, to ensure all participants knew what was meant by these concepts.

**Sociodemographic information.** The TISP survey collected data on many sociodemographic information, of which, in this study, we used gender, age, and level of education.

**Trust in scientists.** In the TISP survey, trust in scientists was assessed using 12 items measuring four established dimensions of trustworthiness, i.e., perceived competence, benevolence, integrity, and openness [22, 35]; hence, three items for each dimension, on 5-point semantic differentials. In this study, we used an index based on these 12 items ($\alpha_{Germany}$ = .93; $\alpha_{South\ Africa}$ = .92).

**Media use.** Respondents were asked to indicate, on a 7-point scale from 1 "never" to 7 "once or more per day", how often they come across information about science in eight different places (e.g., in news articles in printed newspapers or magazines) in the last 12 months. In this study, we categorized the items into "news media" (e.g., news articles, TV or radio, news apps, news podcasts; 4 item; $\alpha_{Germany}$ = .80; $\alpha_{South\ Africa}$ = .87), "social media" (e.g., YouTube, TikTok, WhatsApp; 2 items; $r_{Germany}$ = .58; $r_{South\ Africa}$ = .57), and "non-mediated communication" (e.g., museums, zoos, conversations with friends or family; 2 items; $r_{Germany}$ = .48; $r_{South\ Africa}$ = .50).

**Perceptions and expectations of scientists' visibility.** Respondents were shown a definition for what visible scientists are ("they regularly appear in the mass media including on television, in print and online news, in radio, and on social media platforms") and then explicitly asked: "if you can, please name up to three visible living scientists in your country." In a second open-ended question, they were asked to characterize visible scientists in their own words. Next, we asked them, on five-point scales from 1 "strongly disagree" to 5 "strongly agree", for their perceptions and expectations regarding the ten characteristics of visible scientists introduced before (see Table 1; adapted from 14). Lastly, in another open-ended question, we asked what makes a visible scientist trustworthy.

**Table 1. Perceptions and expectations of visible scientists, comparing German and South African respondents.**

| Variables | As far as I know, visible scientists . . . | | | | Visible scientists should . . . | | | |
|---|---|---|---|---|---|---|---|---|
| | German respondents | | South African respondents | | German respondents | | South African respondents | |
| | *M* | *SD* | *M* | *SD* | *M* | *SD* | *M* | *SD* |
| are/be highly articulate (well-spoken, use accessible language) | 3,52 | 1,032 | 4,05 | 1,024 | 4,14 | ,904 | 4,38 | ,871 |
| have a good scientific reputation. | 3,49 | ,957 | 4,05 | 1,015 | 4,01 | ,943 | 4,56 | ,772 |
| are/be senior. | 3,53 | ,902 | 3,55 | 1,198 | 3,07 | 1,077 | 3,29 | 1,330 |
| are/be media-savvy (work well with the media). | 3,30 | ,992 | 3,57 | 1,185 | 3,52 | 1,006 | 4,01 | 1,015 |
| are/be charismatic (likeable and inspiring leaders). | 3,10 | ,984 | 3,63 | 1,170 | 3,46 | ,966 | 4,05 | 1,133 |
| are/be controversial. | 3,09[a] | ,863 | 3,19[a] | 1,084 | 3,15[a] | ,966 | 3,06[a] | 1,343 |
| comment on topics outside of their expertise. | 2,86 | 1,059 | 3,07[a] | 1,253 | 2,56 | 1,156 | 3,12[a] | 1,463 |
| are/be criticized for seeking the limelight. | 3,00 | 1,032 | 2,83 | 1,279 | 2,30 | 1,181 | 2,36 | 1,395 |
| are/be male. | 3,04 | 1,085 | 2,70 | 1,312 | 2,04 | 1,132 | 2,00 | 1,166 |
| share details about their private lives with the public. | 2,13 | 1,060 | 2,16[a] | 1,207 | 1,87 | 1,128 | 2,11[a] | 1,286 |

*Notes.*

[a] country-comparison between perceptions and expectations is not significant.

### 3.3 Analyzing closed and open-ended questions

Working towards answering RQ1, we started by inductively coding responses to the open-ended question about three visible living scientists in the respective country (e.g., what answers were provided and were names given of scientists who were alive and working in the respective country). We counted all the names of scientists who were alive and worked in the respective country at the time of the survey as valid answers. We then present the quantitative/descriptive data of the standardized items and extend this with inductive coding of further characteristics from the open-ended question asking respondents to characterize visible scientists. To answer RQ2, we inductively coded answers to the open-ended question about what makes visible scientists trustworthy, with a focus on aspects beyond the common dimensions of trustworthiness, and then present correlations between perceptions and expectations of scientists' visibility and trust in scientists, media use, and sociodemographic information. Due to unclear causal paths, we present correlations.

## 4 Results

### 4.1 Public perceptions of visible scientists in Germany and South Africa (RQ1)

In response to the open-ended question "please name up to three visible living scientists in your country", a majority of the survey respondents in both countries did not mention even a single name. In total, 553 German respondents and 545 South African respondents either left the field empty or wrote "I don't know" or a similar response (see Fig 1a and 1b, large circles). Some respondents apologized stating that "I'm sorry, unfortunately I don't know any names of publicly visible scientists"; "Sorry, can't think of any by name"; "Unfortunately, I can't name any names, sad but true." While some blame this fact on themselves: "Unfortunately I can't, remembering names is not my strong point", or "poor memory", several respondents mention that scientists in their country were just not very visible. A German respondent wrote: "I hardly know any by name. I would describe them as invisible. Professor Drosten only became so well-known because of the pandemic". And a South African respondent echoed: "Don't know any of their names. They never appear on TV or talk that much in the media and if they have, I might have missed that moment." Another German respondent stated: "Media presence is not necessarily the sign of good scientific work, which is why I am not mentioning any names here." Yet another reflected: "I can't think of anyone off the top of my head, I could only list my university lecturers, but that seems too random to me and they are not so much in the public eye." Some South African respondents commented on the lack of media exposure for local scientists by saying, for example: "The media doesn't focus on scientists, it is always about politics." Similar examples include: "Funny enough I can't because our media hardly gives them coverage on their expeditions, I, therefore, haven't been exposed to any of them. But I can list a lot of European scientists.", and "In America people like Anthony Fauci and Sanjay Gupta are household names. In South Africa, our scientists don't receive the credit due to them from the media. Does anyone remember the name of the South African scientist who discovered a new variant of the COVID-19 virus?"

447 German respondents and 455 South African respondents attempted to provide at least one name, but many of these were not scientists, or they were scientists living outside the respective countries or deceased scientists, including numerous mentions of Albert Einstein, Marie Curie, Carl Sagan, and Christiaan Barnard. Nicolaus Copernicus, Isaac Newton, Charles Darwin, and Alexander von Humboldt were also mentioned. Three German respondents mentioned virologist Robert Koch, after whom the Robert Koch Institute is named, the

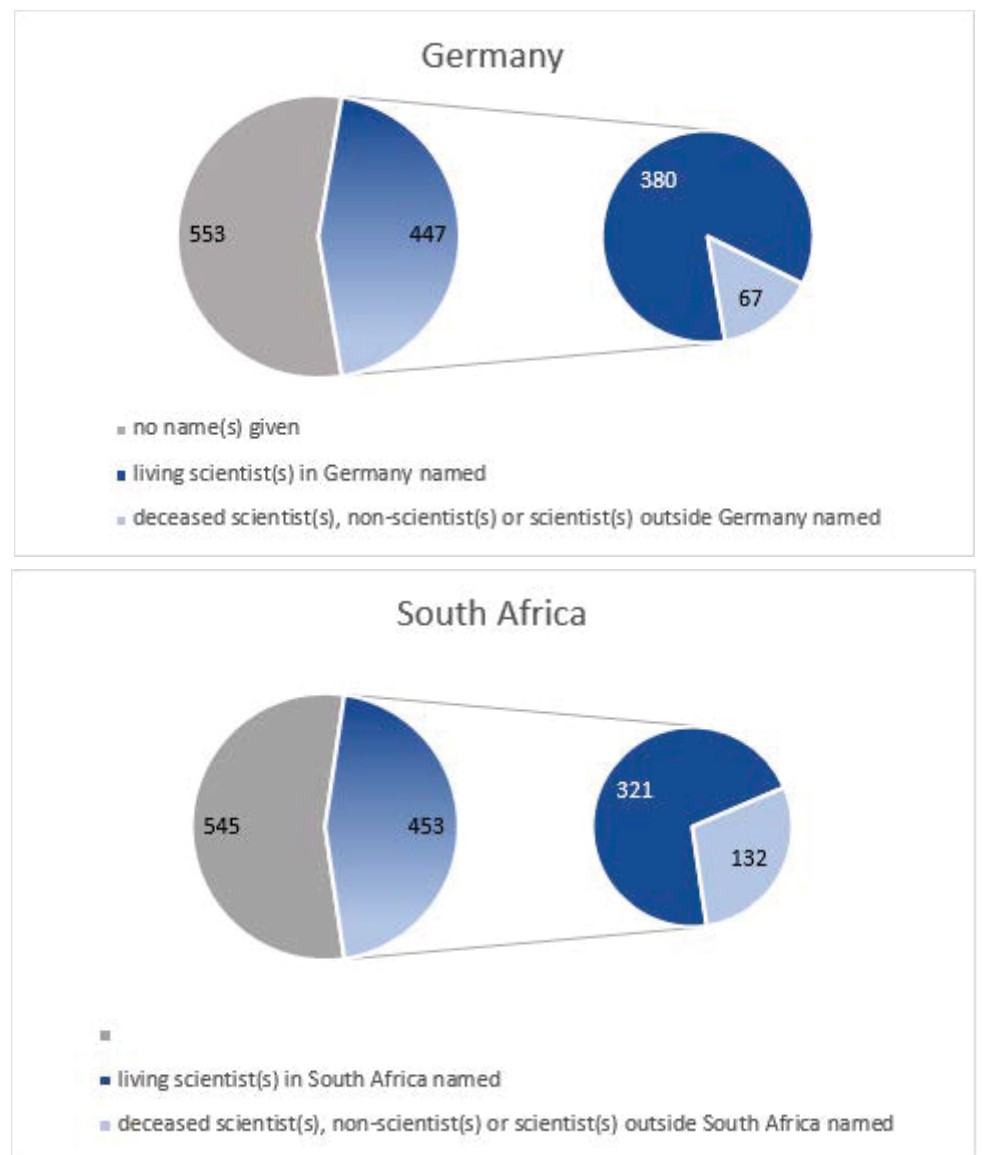

**Fig 1. a.** Responses to the question "please name up to three visible living scientists in your country" in Germany. **b.** Responses to the question "please name up to three visible living scientists in your country" in South Africa.

German federal government agency responsible for disease control and prevention, which played a major role in Germany's COVID-19 communication. Among the non-scientists mentioned were politicians, journalists, and science communicators. In Germany, 82 respondents named Karl Lauterbach, the minister of health in office at the time of the survey and who also holds academic degrees, and about 40 respondents named science communicators such as Mai Thi Nguyen-Kim, Harald Lesch (who is also a scientist), Ranga Yogeshwar, and Eckart von Hirschhausen. Names in South Africa included Jacob Zuma (former president), Zweli Mkhize (former health minister), and science writers Daryl Ilbury, Lia Labuschagne, and Lynne Smit. 15 South African respondents named tech entrepreneur Mark Shuttleworth, who became famous in 2002 for becoming the first citizen of an independent African country to go into space, traveling with the Russian Soyuz TM-34 mission as a space tourist, and who has been

living in the United Kingdom for many years. Elon Musk, a businessman and entrepreneur who left South Africa in 1988, was mentioned 14 times.

Of the respondents who attempted to give a name, 380 in Germany and 321 in South Africa were able to mention at least one scientist whom they perceive to be visible and who is alive and works in their respective countries (Fig 1a and 1b, small circles). Looking more closely at the names given, we find that in the 321 South African responses, 134 unique visible scientists were mentioned, of which 74 were female and 60 were male. While in South Africa, we find a majority of female names mentioned, the quota of female names given is much lower in Germany. Respondents in Germany mentioned 140 unique names, of which 102 were male and only 38 were female.

Regarding disciplines, virology dominated the German data by far. 174 of the 380 valid responses named Christian Drosten, and thus confirmed him as the German "Pandem-icon" [12]. The next most-frequently mentioned scientists were also visible in relation to COVID-19 (47 mentions for virologist Hendrick Streeck and 27 for Lothar Wiehler, the chief executive officer of the Robert Koch Institute), leaving climate scientists (29 mentions for Mojib Latif), and disciplines such as physics, medicine, economy, and social sciences less able to compete. The names given are thus closely tied to public issues. An exception is the winners of the Nobel Prize. Mentions included Emmanuelle Charpentier, Harald zur Hausen, Christiane Nüsslein-Vollhard as well as several living and deceased Nobel prize-winning physicists.

In South Africa, HIV experts were frequently mentioned, for example 42 mentions for Linda Gail Bekker, 32 for Salim Abdool Karim, and eight for Glenda Gray. It should be noted that Abdool Karim and Gray were frequently quoted in the media during the COVID-19 pandemic. Chemist and cancer researcher Tebello Nyokong was mentioned 56 times, and palaeontologist Lee Berger, who invoked some controversy around his findings of human ancestors, was mentioned 44 times. Mentions of climate scientists were negligible: there were only six mentions for Bruce Hewitson and only two for Guy Midgley, both internationally renowned climate scientists who live and work in South Africa.

Frequently mentioned, however, was Wouter Basson (73 mentions), a highly controversial cardiologist and former head of South Africa's secret chemical and biological warfare project, Project Coast. During the apartheid era, Wouter Basson was given the nickname "Dr. Death" in the media for his alleged role in the deaths of anti-apartheid activists [36, 37]. In 2002, he was acquitted of 67 charges related to his involvement in apartheid-era crimes, an acquittal that has been regarded as hugely controversial [38]. A public outcry erupted when it emerged in 2021 that he had been practising as a cardiologist at a local private hospital since 2005.

Looking at the standardized items across respondents from both nations, visible scientists were seen as highly articulate and with a good scientific reputation. Most respondents agreed that they are senior, media-savvy, charismatic, and even controversial. Respondents seemed more undecided if visible scientists comment on topics outside their expertise, are criticized for seeking the limelight, and are male. They tended to disagree with the statement that visible scientists share details about their private lives with the public (Table 1).

Some of these perceptions differed between German and South African respondents (see Fig 2). Most respondents agreed that visible scientists are highly articulate, with South African respondents agreeing more with this than German respondents ($t = 11.520$; $df = 1.996$; $p < .001$). South African respondents also tended to agree more that visible scientists have a good scientific reputation ($t = 12.719$; $df = 1.996$; $p < .001$), are media-savvy ($t = 5.445$; $df = 1.996$; $p < .001$), charismatic ($t = 10.936$; $df = 1.933$; $p < .001$), controversial ($t = 2.273$; $df = 1.902$; $p < .05$), and comment on topics outside of their expertise ($t = 5.445$; $df = 1.996$; $p < .001$). In contrast, German respondents tended to agree more that visible scientists are criticized for seeking the limelight ($t = 3.292$; $df = 1.909$; $p < .001$) and are male ($t = 6.310$; $df = 1.928$; $p < .001$).

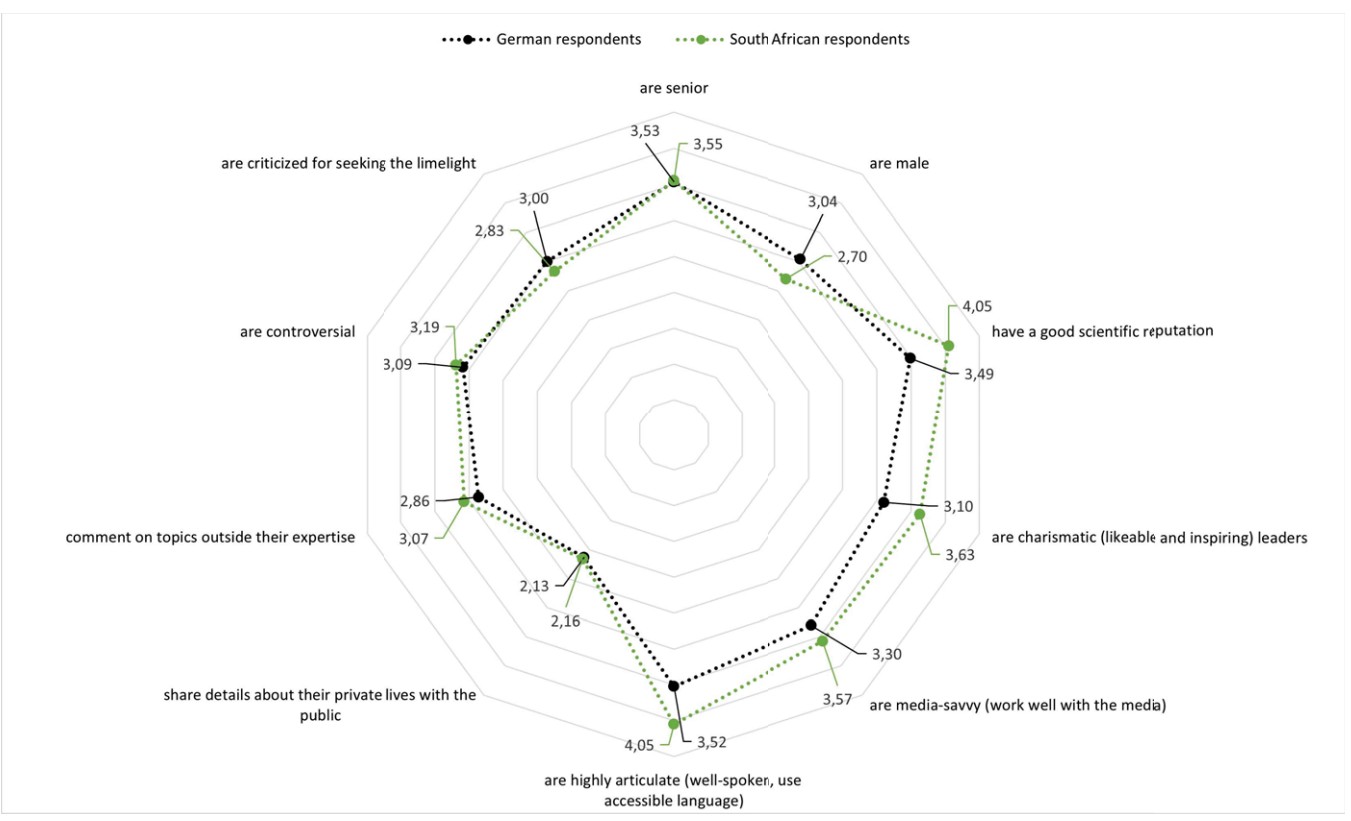

**Fig 2. How German and South African respondents perceive visible scientists.** (Items: "As far as I know, visible scientists . . .").

There were no differences in the samples regarding visible scientists' seniority and sharing details about their private lives with the public.

When it comes to expectations, the ranking is similar to the perceptions. Expectations for visible scientists to have a good scientific reputation are slightly higher than the expectation for them to be highly articulate. People also expect them to be media-savvy and charismatic; there is some indifference regarding seniority and controversy. However, respondents did not expect visible scientists to comment on topics outside of their expertise, to be criticized for seeking the limelight, and to be male. The most disagreed upon expectation is the one that visible scientists should share details about their private lives with the public (see again Table 1).

In the same vein, some expectations of visible scientists differed between German and South African respondents (see Fig 3). Most respondents agreed that visible scientists should have a good scientific reputation, again with South African respondents agreeing more with this than German respondents ($t = 14.075$; $df = 1.920$; $p < .001$). They also tended to agree more that visible scientists should be highly articulate ($t = 5.973$; $df = 1.996$; $p < .001$), media-savvy ($t = 10.916$; $df = 1.996$; $p < .001$), charismatic ($t = 12.335$; $df = 1.948$; $p < .001$), senior ($t = 4.114$; $df = 1.914$; $p < .001$), should comment on topics outside of their expertise ($t = 9.545$; $df = 1.896$; $p < .001$), and should share details about their private lives with the public ($t = 4.326$; df = 1.957; $p < .001$). German respondents tended to agree more that visible scientists should be controversial ($t = 1.812$; $df = 1.810$; $p < .05$). There were no differences in the samples regarding expectations that visible scientists should be criticized for seeking the limelight and should be male.

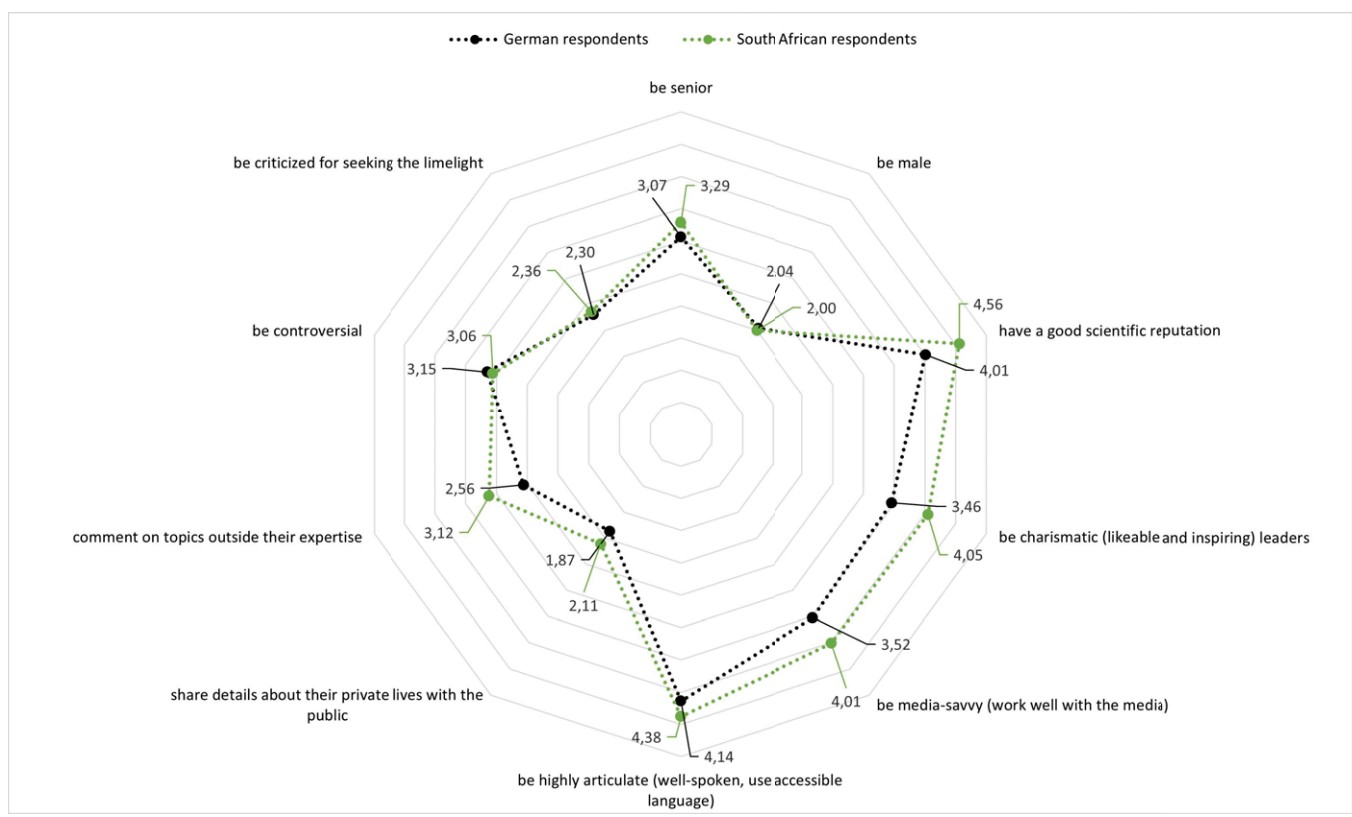

**Fig 3. How German and South African respondents think visible scientists should be.** (Items: "Visible scientists should...").

Setting perceptions and expectations in direct comparison (see Table 1) allows one to assess where respondents saw the biggest discrepancies between perceptions and expectations. Both German and South African respondents were quite similar in this respect. Indifferent assessments pertain to the sharing of private details, commenting on topics outside one's expertise, and for being controversial. What respondents expected less was for visible scientists to be male, to be criticized for seeking the limelight, and to be senior. What respondents expected more was for visible scientists to be media-savvy, to be charismatic leaders, to have a good scientific reputation, and to be highly articulate.

Inductive coding of further characteristics from the second open-ended question, "If you think about visible scientists, how would you describe and characterize them", demonstrated that the ten characteristics analyzed above cover most of the aspects that respondents commented on in their own words. Namely articulate, intelligent, hard-working, dedicated, honest, charismatic, and leadership came up in their answers. When we explicitly looked for what else was mentioned that was not covered by the ten criteria, respondents raised affirmative as well as critical points.

In Germany, visible scientists were pointedly characterized as "people who are interviewed in front of bookshelves" who are in the media "because they work on public issues". Affirmative answers included that they are curious, passionate, down-to-earth, confident, and sympathetic. They were also described as taking on important real-world problems and contributing to solving them with their discoveries. Among the open-ended answers, however, critical stances prevailed, including features such as arrogant, cold, peculiar, socially incompetent, alarmist, not honest, not disinterested, too political, and/or too closely intertwined with politics and, therefore, not independent.

Affirmative answers in South Africa included that visible scientists "put the needs of others before their own", that they are "geniuses that are impulsive and have a lot of creativity", and that they "bring a concept of ubuntu in science", an ancient African word meaning ´humanity to others´ and a reminder of that people are what they are because of others (Source: https://ubuntu.com/about). Critical remarks included the following: "weird and nerdy", "love the limelight and think they are superior to other people", "god-complex", and "money grabbers". As in Germany, several respondents criticized too close ties to politics, or a lack of independence: "They could have done better if they were not in the hands of politicians", "they are paid by the government and cooperate. They have the facts and the solutions but there are others at play who are protecting their profits and interests." Eventually, a few respondents explained that they perceive the role of the visible scientist as ambivalent, or even fake: "They are fake scientists, real scientists do not appear in public!". A German respondent stated: "When you see them too often in the media, I often get the feeling that they are only doing it to achieve 15 minutes of fame. Genuine and credible scientists tend not to need this and do their work more in the background." A South African respondent seconds: "The idea of visible scientists itself is a bit repulsive. You're in it for the love of science or for the glory. Visible scientists seem like self-serving narcissists."

## 4.2 The link between perceptions and expectations of visible scientists, trust in scientists, and media use (RQ2)

In response to the question *"What do you think, what makes a visible scientist trustworthy*?*"*, we found that the respondents' open-ended statements mostly agree with the established trust dimensions of expertise, integrity, benevolence, transparency, and dialogue [20]. This includes several instances where the lack of any of these dimensions was given as a cause of distrust in visible scientists. Repeatedly, respondents expressed that they suspect visible scientists not to tell the truth for political reasons: "I do not trust them at all because they usually push government agendas", "they are too close to the *Zeitgeist*", or "I have given up on the subject. ONLY those scientists who are NEVER associated with politics or politicians are trustworthy for me and who do NOT talk at the mouth of politicians, but bluntly tell the truth even if the politicians or the paid media do NOT like it!!!!!" (original emphasis). Others stated: "They must tell the truth even if it is inconvenient", and "Their credentials seem fabricated for the sake of legitimacy. I believe that a panel of scientists would be much more trustworthy, because you can fool the average person, but you cannot fool your peers."

In a few answers, it was explicitly the visibility that was given as cause for distrust. Statements from South Africa include: "Unlikely to be trustworthy, seeking limelight instead of doing research", and "I don't think being visible makes you trustworthy. On the contrary, I feel that the less notoriety, the more honest. It is more the work being done that should be visible. Certainly, the person doing the work should be acknowledged and credited but the focus should always be on the work done." The respondents also identified additional factors that would instill trust in scientists, including that everything they say is evidence-based (in both countries), that they lead by example (in both countries), that they are religious (in South Africa), that they openly acknowledge uncertainties and non-knowledge (in Germany) and that they do not act as know-it-alls (in Germany). We did not ask for trust in other professions, but there were indications that the substance of their claims makes scientists in principle more trustworthy than others: "In contrast to journalists and politicians, I hope that scientists know what they are talking about and recognize facts."

Turning to the quantitative data, for the German sample (see Table 2), we find statistically meaningful correlations between several characteristics of visibility and trust in scientists–this

**Table 2. Correlations between perceived characteristics of visible scientists and indicators of trust and media use in Germany.**

| | Perceptions of visible scientists (items: "As far as I know, visible scientists …") | | | | | | | | | |
|---|---|---|---|---|---|---|---|---|---|---|
| | are senior | are male | have a good scientific reputation | are charismatic | are media-savvy | are highly articulate | share details about their private lives | comment on topics outside their expertise | are controversial | are criticized for seeking the limelight |
| Trust in scientists | ,224*** | ,139*** | **,504*** | **,530*** | **,337*** | **,434*** | ,184*** | -0,001 | ,266*** | -0,029 |
| Use of news media | ,133*** | ,063* | ,156*** | ,156*** | ,191*** | ,122*** | ,125*** | ,113*** | ,206*** | ,092** |
| Use of social media | 0,052 | ,095** | -0,019 | ,066* | ,081* | 0,020 | ,248*** | ,200*** | ,115*** | ,081* |
| Use of non-mediated communication | ,095** | ,070* | ,071* | ,173*** | ,184*** | ,101** | ,241*** | ,178*** | ,171*** | ,094** |
| Gender (1 = male) | 0,028 | 0,007 | 0,009 | 0,044 | 0,021 | -0,004 | ,097*** | ,131*** | 0,043 | ,064* |
| Age | ,101** | -,136*** | ,089** | 0,036 | -0,001 | 0,015 | -,183*** | -,110*** | 0,053 | -0,006 |
| Education | 0,050 | 0,058 | ,130*** | ,157*** | ,123*** | ,131*** | ,079* | 0,055 | 0,046 | 0,009 |

*Notes.*

\* p < .05

\*\* p < .01

\*\*\* p < .001. Correlations >.3 in bold.

is especially true for the perceptions that visible scientists have a good scientific reputation, are charismatic, and articulate. There is no correlation for commenting on topics outside the area of expertise and for being criticized for seeking the limelight. The use of news media and non-mediated communication correlated positively with all characteristics of scientists' visibility–the use of social media only with some of them. In comparison, sociodemographic information only shows selected and weak correlations.

For the South African sample (see Table 3), we also find meaningful correlations between several characteristics of visibility and trust in scientists, especially for perceptions that visible scientists have a good scientific reputation, are charismatic, and articulate. We find weaker and negative correlations for the perceptions that visible scientists are male and criticized for

**Table 3. Correlations between perceived characteristics of visible scientists and indicators of trust and media use in South Africa.**

| | Perceptions of visible scientists (items: "As far as I know, visible scientists …") | | | | | | | | | |
|---|---|---|---|---|---|---|---|---|---|---|
| | are senior | are male | have a good scientific reputation | are charismatic | are media-savvy | are highly articulate | share details about their private lives | comment on topics outside their expertise | are controversial | are criticized for seeking the limelight |
| Trust in scientists | ,144*** | -,111*** | **,458*** | **,390*** | ,230*** | **,381*** | 0,051 | 0,021 | 0,011 | -,135*** |
| Use of news media | 0,057 | -0,019 | ,128*** | ,141*** | ,128*** | ,091** | ,101** | ,120*** | 0,053 | 0,062 |
| Use of social media | 0,056 | 0,000 | ,074* | ,132*** | ,098** | 0,024 | ,144*** | ,124*** | 0,061 | ,073* |
| Use of non-mediated communication | 0,046 | 0,041 | ,073* | ,148*** | ,073* | 0,019 | ,094** | ,109** | ,067* | ,078* |
| Gender (1 = male) | 0,000 | 0,001 | -0,047 | -,122*** | 0,002 | -,076* | ,101** | 0,014 | -0,003 | 0,056 |
| Age | -,112*** | -,184*** | -0,032 | -,097** | 0,005 | -,072* | -,094** | -0,053 | -0,032 | -,170*** |
| Education | 0,063* | 0,040 | 0,047 | 0,043 | 0,028 | 0,030 | 0,028 | 0,008 | 0,017 | 0,018 |

*Notes.*

\* p < .05

\*\* p < .01

\*\*\* p < .001. Correlations >.3 in bold.

**Table 4. Correlations between expected characteristics of visible scientists and indicators of trust and media use in Germany.**

| | Expectations of visible scientists (items: "Visible scientists should...") | | | | | | | | | |
|---|---|---|---|---|---|---|---|---|---|---|
| | be senior | be male | have a good scientific reputation | be charismatic | be media-savvy | be highly articulate | share details about their private lives | comment on topics outside their expertise | be controversial | be criticized for seeking the limelight |
| Trust in scientists | ,276*** | ,097** | ,227*** | ,248*** | **,317*** | ,166*** | ,119*** | ,110*** | ,072* | -,078* |
| Use of news media | ,134*** | ,106** | ,199*** | ,133*** | ,126*** | ,111*** | ,132*** | ,065* | ,110** | 0,045 |
| Use of social media | ,122*** | ,218*** | 0,006 | 0,048 | 0,015 | -0,053 | ,277*** | ,119*** | ,097** | ,089** |
| Use of non-mediated communication | ,154*** | ,193*** | ,129*** | 0,060 | ,072* | 0,038 | ,262*** | ,138*** | ,147*** | ,081* |
| Gender (1 = male) | 0,054 | ,133*** | ,124*** | 0,016 | 0,053 | -0,056 | 0,058 | 0,038 | ,116*** | ,177*** |
| Age | 0,003 | -,208*** | ,128*** | 0,042 | 0,034 | ,141*** | -,197*** | 0,006 | ,088** | 0,008 |
| Education | -0,004 | 0,053 | ,076* | 0,048 | 0,057 | 0,043 | 0,041 | -0,037 | -0,022 | -0,044 |

*Notes.*

* $p < .05$

** $p < .01$

*** $p < .001$. Correlations $>.3$ in bold.

seeking the limelight. In the South African sample, the use of news media, social media, and non-mediated communication did correlate with perceiving them as having a good scientific reputation, being charismatic, being media-savvy, sharing details about their private lives, and commenting on topics outside the area of expertise. Again, sociodemographic information only shows selected and weak correlations.

When it comes to expectations, in the German sample (see Table 4), we see weaker correlations between characteristics of scientists' visibility and trust in scientists–the strongest positive correlation emerged for the expectation that visible scientists should be media-savvy. A possible interpretation is that trust in scientists is linked more strongly to perceptions than expectations. Noteworthy are the weak correlations for the expectation that visible scientists should be male, controversial, and the negative ones for that they should be criticized for seeking the limelight. Hence, respondents with higher trust in science see these aspects as less relevant or controversial, or else, the less trust the more it is expected that visible scientists should be criticized for seeking limelight. Patterns for media use and the weak correlations for sociodemographic information are similar to the perceptions, with only small differences.

We find similar patterns with only some differences for perceptions and expectations for the South African sample (see Table 5)–this is true for use of news media and social media, non-mediated communication, and sociodemographic information. As with the perceptions, correlations emerged between trust in scientists and expectations for visible scientists to have a good scientific reputation, for being charismatic, media-savvy, and articulate. Again, there were weaker and negative correlations for the expectations for visible scientists to be male and to be criticized for seeking the limelight.

## 5 Discussion

This paper contributes to exploring the social role of visible scientists by studying how publics in Germany and South Africa view visible scientists. This implies a double interest in how the public perceives individuals who play the role of visible scientists, as well as an interest in the expectations that these publics have towards scientists who become visible, and therefore take on a specific social role. To this end, we combined closed questions for characteristics taken

**Table 5. Correlations between expected characteristics of visible scientists and indicators of trust and media use in South Africa.**

| | Expectations of visible scientists (items: "Visible scientists should...") | | | | | | | | | |
|---|---|---|---|---|---|---|---|---|---|---|
| | be senior | be male | have a good scientific reputation | be charismatic | be media–savvy | be highly articulate | share details about their private lives | comment on topics outside their expertise | be controversial | be criticized for seeking the limelight |
| Trust in scientists | -0,002 | -0,053 | ,184*** | ,183*** | ,203*** | ,181*** | -0,056 | -0,050 | 0,048 | -,111*** |
| Use of news media | ,081* | 0,046 | 0,003 | ,090** | ,142*** | ,063* | ,105** | ,198*** | ,075* | 0,052 |
| Use of social media | ,146*** | ,083** | -0,031 | ,124*** | 0,056 | 0,034 | ,218*** | ,188*** | ,080* | ,093** |
| Use of non-mediated communication | ,188*** | ,089** | ,082** | ,151*** | ,067* | ,078* | ,205*** | ,208*** | ,068* | 0,037 |
| Gender (1 = male) | ,135*** | ,127*** | 0,017 | -,071* | 0,006 | -0,034 | ,068* | ,063* | -,151*** | ,111*** |
| Age | -0,060 | -,173*** | 0,016 | -,106** | -0,029 | -,097** | -,117*** | -,163*** | -0,049 | 0,005 |
| Education | 0,008 | 0,014 | 0,047 | 0,021 | -0,004 | 0,014 | -0,011 | -0,021 | -0,014 | -0,006 |

*Notes.*

* p < .05

** p < .01

*** p < .001. Correlations >.3 in bold.

from the literature with open-ended questions, which the respondents could answer in their own words.

Non-responses to the request to name up to three visible scientists living in the respective countries demonstrate that overall, scientists are invisible rather than visible in public, and that the visible scientist is–and remains–a rare phenomenon despite changing media environments and a recent global pandemic. A majority of the public in both countries could not give a single name, and, despite our explicit request to name living scientists, there were many deceased scientists among the names mentioned. The iconic Einstein, Sagan, and Barnard featured prominently.

The actual names of living scientists that were given reveal how closely the responses link to recent public issues. Christian Drosten, Germany's public face in the COVID-19 pandemic, accounts for more than half of the overall mentions in Germany. Other virologists follow suit, leaving climate scientists and scientists from other disciplines less able to compete. Notably, this picture looks different in South Africa. While scientists who spoke out on COVID-19, such as Salim Abdool Karim, were mentioned, recent renewed controversies around a controversial apartheid-era doctor, Wouter Basson, catapulted him to the top of the South African mentions. Our findings confirm that other factors, rather than the pandemic only, influenced the visibility of scientists in the South African media. This resonates with earlier findings showing that while COVID-19 generally dominated media headlines around the world during the early stage of the pandemic, it was not equally dominant across countries and media coverage varied depending on local contexts [39]. A study of how newspaper cartoonists featured COVID-19 from 1 January 2020 to 30 June 2021 [40] showed that 66% of editorial cartoons in the UK were about COVID-19, compared to only 36% in South Africa, confirming that there were many competing news stories in South Africa. This study also highlights the media relevance of poverty, corruption, and crime in South Africa.

We conclude that the names given are closely tied to news cycles, with scientists working on public issues such as the pandemic and climate change mentioned most often, along with scientists that are controversial for their own behavior. A notable exception is winners of the Nobel Prize, confirming the literature that has established that this academic elite status also

nominates for media prominence. Interestingly, several respondents also named politicians (especially ones who were related to the COVID-19 pandemic) and science communicators. This might be taken as an indicator that whoever talks about science in public is easily perceived to be a scientist.

In the quantitative data, the key characteristics in the public's perception correspond to the combination of scientific and personal attributes that Goodell summed up as articulate, colorful, controversial, and reputable as a scientist, and which virtually any case study on visible scientists since has confirmed. From German and South African respondents' point of view, the most relevant perceptions see visible scientists as highly articulate, media-savvy, and charismatic, with a sound scientific reputation. and a certain degree of seniority. Except for seniority, these characteristics also feature prominently in the respondents' expectations, with scientific reputation being the most expected criterion. It is thus highly likely that the credibility of visible scientists is predominantly linked to their scientific reputation. Of interest in this regard is how these public perceptions relate to a recent study [13], which traced the trajectories of scientists from entering the media arena to becoming visible scientists, and which found that, according to their h-indexes, some of these scientists had low scientific reputation.

Comparing perceptions and expectations between both samples showed that respondents want visible scientists to be media-savvy, to be charismatic leaders, to have a good scientific reputation, and to be highly articulate, again in line with the literature. In contrast, male dominance, critique for seeking the limelight, and seniority are characteristics that the respondents would prefer to be less prevalent in the future. The difference between perceived and expected seniority might give reason to efforts to feature more junior and more female scientists in the public sphere. In both countries, respondents are not interested in details about scientists' private lives. These findings can give (science) journalists an idea of what publics expect from the visible scientists in the media. To focus on their professional role and not feature details about their private lives may also serve as a recommendation for scientists' self-mediation, for instance on social media.

Concerning RQ2, links between media use, trust in science, and perceptions as well as expectations, there were also similar patterns across the samples and across the "is" and the "should". The more respondents trusted scientists, the more they think/expect visible scientists (to) have a good scientific reputation, be charismatic, be articulate, and be media-savvy. The use of media, especially news media, seems to stand in a (although not strong) relationship with several perceptions and expectations of scientists' visibility. We find stronger connections for perceptions (compared to expectations) and for the trust item (compared to the use of media items). This shows, to some degree, that although media use must be a strong factor for publics to get in contact with science, the frequency of media use stands nevertheless in a weaker connection to perceptions of/expectations towards visible scientists. Trust in scientists, however, seems to stand in a stronger relationship.

The qualitative as well as the quantitative findings show only minor differences between Germany and South Africa. In contrast to the specific and very up-to-date names given, this indicates that the general perceptions of and expectations towards the social role of the visible scientist are rather stable and similar for these two countries from the Global North and the Global South, respectively. The overall similar attitudes towards visible scientists may be explained by a universal public image of science around the world, which includes expectations towards scientists as visible scientists. This finding is confirmed beyond the two cases in another recent paper that studied 16 countries and reports similar characteristics for becoming a visible scientist [12]. While similar patterns emerge overall, South African respondents showed a tendency to agree more than German respondents, for both the perceptions and the expectations–a finding that would need further inquiry to be better understood.

How can we interpret that most respondents in both countries did not provide a single name of a visible scientist who currently lives in their country? Goodell [2] already suggested, based on her qualitative interviews with scientists, that visible scientists are exceptions. Several studies have confirmed this proposition and shown that only a small number of scientists speak out repeatedly in public, whereas the majority of scientists rarely speak out in the media [41]. In a review of five surveys of German scientists with a total of over 1.700 respondents, Peters [42] comes to a similar result: experiences with the media are "common among scientists, but not routine for most." It is thus not the addressing of non-scientific audiences per se which is rare but the repeated media presence of individual visible scientists. For the first time, our findings now shed light on this from a different perspective, the perspective of publics in Germany and South Africa, and confirm exactly this point. The names mentioned include the iconic examples of past centuries–Copernicus, Newton, Darwin, Curie, Einstein, Sagan, and Barnard–as well as the names of relatively few men and even fewer women, whose expertise is currently in demand in relation to recent public issues such as COVID-19 and climate change. This was confirmed in the qualitative data in which visible scientists were repeatedly referred to as a "temporary phenomenon". The German data show the dominance of COVID-19 related scientists being named with a single scientist getting almost half of the overall mentions. Interestingly, the South African data show a different pattern, in which COVID-19 related public visibility played a less prominent role and respondents focused on scientists surrounded by controversy (such as Wouter Basson and Lee Berger) or those working in the field of HIV/Aids, but who were also prominent during COVID-19 (such as Salim Abdool Karim, Linda Gail Bekker, and Glenda Gray).

Another part of the explanation is related to the ambivalent relationship of peer communities with their visible colleagues. Goodell [2] established that "visible scientists are seen by their colleagues almost as a pollution in the scientific community–sometimes irritating, sometimes hazardous." Rödder [4] has pointed to the scientific community's fear of losing peer control and reputation autonomy over visible scientists. While the quantitative data indicate that visible scientists should be less criticized for seeking the limelight than they actually are, it is noteworthy that the qualitative data mirror this ambivalence for the surveyed publics. Concerns about self-promotion, arrogance, and narcissism were named as relevant reasons why you cannot trust scientists, and even contradictory positionings of reputation and prominence were mentioned: "The real ones, you do not hear about." Furthermore, Fiske and Dupree's [26] finding of ambivalent emotions towards scientists in general, which includes both admiration and resentment, can be confirmed for the visible scientists we asked about in this study. While there were many affirmative expressions, the trust dimensions were also mentioned as often lacking in visible scientists, namely independence, honesty, disinterestedness, accessibility, and thus trustworthiness. The public's expectations are clear from Fig 3, and from the point of view of some respondents, there seems to be a conflict with these role expectations.

Eventually, what Goodell has described as a "comfortable symbiosis" [2] between scientists and journalists, is perceived by the public as a threat to the independence of both research and journalism. This is clearly shown in the qualitative data: scientists should not have any ties to media, lobby groups, or politics, and should not speak out in favor of a certain political position or policy. Given examples mainly from the COVID-19 pandemic, some respondents felt that an individual scientist's perspective shaped media reporting as well as political decision-making to a large, for some unduly large, extent. The phenomenon of "anything authority", i.e., the tendency for scientists to comment outside their field of expertise, is particularly frowned upon. Following up on the interesting suggestion, that "a panel of scientists would be much more trustworthy, because you can fool the average person but you cannot fool your peers", future studies should compare the role of individual scientists with organizational

COVID-19 communication efforts such as by Science Media Centers [43] to shed light on this point further.

Before we come to our conclusions, a few limitations of this study need to be pointed out. First, we study Germany as a country from the Global North and South Africa as a country from the Global South, mainly to explore and understand their similarities and differences. We do not claim that either country is 'representative' of the respective geopolitical category and future studies should enlarge their samples to include more countries from across the world. An interesting finding and potential implication in terms of the Global North and the Global South is that the COVID-19 pandemic played much more of a role as a single public issue in our Global North case than in our Global South case. This finding could be followed up by studies of all sorts of crisis news issues across countries A second limitation is the cross-sectional nature of our study, which should be complemented by panel surveys in the future. Additional content analyses to measure the degree to which the names given for visible scientists actually appear in journalistic media (or are active on social media, including their follower counts) would also complement this research.

## 6 Conclusions

Following Goodell's initial work and label of "visible scientist", there has been a debate about the appropriate terminology, and the term "celebrity" has also been proposed [3]. While Fahy and Lewenstein [6] note that the digitalization and increase of media platforms make celebrity an omnipresent feature of contemporary culture, they single out Carl Sagan as a case in point. While we agree that celebrity (or media prominence) is undoubtedly an omnipresent feature in the 21st century, we have shown that publics in two countries across the globe in their majority cannot name a single scientist living and working in their country. Furthermore, the quantitative data clearly indicate that German and South African respondents are not interested in detail about scientists' private lives, in stark contrast to other celebrities such as the Kardashians or the Royals. We thus doubt that celebrity is an appropriate term to use in relation to scientists. Sagan is a case from the 20th century, rather than the 21st, and that he has still to be cited as a case in point is indicative of how few of these prominent scientists there are.

Overall, our data clearly indicates that visible scientists are mostly seen as scientists who appear in the news when their topics hit the headlines as a public issue. For the vast majority of scientists in the public eye therefore, visibility is temporary. For the average scientist, it is when their topics become public issues that they become visible. We thus suggest that "visibility" is the more appropriate term to refer to this rather temporary phenomenon. These findings and conclusions imply that data such as from this survey are deeply historical, and that–beyond Einstein, Sagan, and Barnard–a high fluctuation and contingency of names given in subsequent surveys and other studies is to be expected.

## Supporting information

**S1 File. Inclusivity in global research.**
(DOCX)

## Acknowledgments

This study is part of the crowd-sourced Many Labs project "Trust in Science and Science-Related Populism" (TISP), in which the authors collaborated with researchers around the world (for a project overview, see [33], for the data set [34]). We gratefully acknowledge the

project heads Viktoria Cologna and Niels Mede for initiating the TISP project and setting up the basic infrastructure. We thank our student research assistants Linda Winkler for quality control of the data analysis, and Charlotte Massarrat-Maschhadi for support in the preparation of the manuscript.

## Author Contributions

**Conceptualization:** Simone Rödder, Lars Guenther, Marina Joubert.

**Formal analysis:** Simone Rödder, Lars Guenther, Marina Joubert.

**Funding acquisition:** Simone Rödder.

**Methodology:** Lars Guenther.

**Software:** Lars Guenther.

**Writing – original draft:** Simone Rödder.

**Writing – review & editing:** Simone Rödder, Lars Guenther, Marina Joubert.

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
