## [Decision Letter · Decision Letter 0]

6 Sep 2024

PONE-D-24-24028“They never appear on TV and if they have, I might have missed that moment.” How publics in South Africa and Germany view visible scientistsPLOS ONE

Dear Dr. Rödder,

Thank you for submitting your manuscript to PLOS ONE. After careful consideration, we feel that it has merit but does not fully meet PLOS ONE’s publication criteria as it currently stands. Therefore, we invite you to submit a revised version of the manuscript that addresses the points raised during the review process.

We look forward to receiving your revised manuscript.

Kind regards,

Emerald Jay Ilac

Academic Editor

PLOS ONE

[No]. 

Additional Editor Comments (if provided):

Reviewers' comments:

Reviewer's Responses to Questions

**Comments to the Author**

1. Is the manuscript technically sound, and do the data support the conclusions?

Reviewer #1: Yes

Reviewer #2: Yes

2. Has the statistical analysis been performed appropriately and rigorously? 

Reviewer #1: Yes

Reviewer #2: Yes

3. Have the authors made all data underlying the findings in their manuscript fully available?

Reviewer #1: Yes

Reviewer #2: Yes

4. Is the manuscript presented in an intelligible fashion and written in standard English?

Reviewer #1: Yes

Reviewer #2: Yes

5. Review Comments to the Author

Reviewer #1: Thank you for allowing me to review your manuscript. The topic was very interesting. A few suggestions that I hope will help:

- Given that this is a cross-country study, I appreciated the section in the introduction that explained the perceptions and expectations in Germany and South Africa. In line with this, the discussion could be improved by expounding further on the implications of the results in terms of the Global North and South. Also, are there any cultural factors that should be noted?

- What are the limitations of the study?

- Some figures appear to be blurry.

Reviewer #2: The study presents relevant findings. Given how many "experts" speak or post their opinions/findings in social media, it offers insight as to how the public is likely to perceive credible scientists who become visible. Because the study compared and contrasted perceptions from 2 countries, it may be best to organize the data using tables where data coming from South Africa is on one side/column and data from Germany is on the other side/column so that similarities and differences are easier to track. The existing tables containing numbers are helpful but these can be confusing to understand especially if the reader would like to focus on qualitative information.

It may also help if authors discuss the theoretical/practical implications of the research. Can future studies focus on the current study's limitations? How can the findings guide scientists who wish to become visible or are asked to speak in public about their research and/or findings? How do they maintain credibility given the public's perception of visible scientists?

6. PLOS authors have the option to publish the peer review history of their article (what does this mean?). If published, this will include your full peer review and any attached files.

Reviewer #1: No

Reviewer #2: No

---

## [Author Response · Author response to Decision Letter 0]

27 Sep 2024

Reviewer comment Authors’ response

Reviewer #1 

Given that this is a cross-country study, I appreciated the section in the introduction that explained the perceptions and expectations in Germany and South Africa. In line with this, the discussion could be improved by expounding further on the implications of the results in terms of the Global North and South. Also, are there any cultural factors that should be noted? Thank you for this important comment. In addition to the explanations in the previous version of the manuscript, we have now made it clearer (p. 5), that – while we study one country from the Global North and one country from the Global South – we do not necessarily see Germany and SA as ‘representative’ of their respective geopolitical category (although we do acknowledge the notable socio-economic differences between these two countries). As regards the cultural factors, the argument then is that our result of an overall rather similar perception of visible scientists in one Global North and one Global South country may be explained by a universal public image of science around the world, which also includes expectations towards scientists as visible scientists. This finding is confirmed beyond the two cases in another recent paper that studied 16 countries and reports rather similar characteristics for becoming a visible scientist across countries (Joubert et al. 2023). 

What are the limitations of the study? Thanks for this comment. We have added a paragraph on the limitations of our study on p. 31.

Some figures appear to be blurry. Thank you for pointing this out. This seems to be an unfortunate effect of scaling up the figure size in the proof manuscript. The original Fig. 1a and 1b are not blurry and they should reproduce with acceptable quality if inserted at the correct (intended) size in the final proof.

Reviewer #2 

Because the study compared and contrasted perceptions from 2 countries, it may be best to organize the data using tables where data coming from South Africa is on one side/column and data from Germany is on the other side/column so that similarities and differences are easier to track. The existing tables containing numbers are helpful but these can be confusing to understand especially if the reader would like to focus on qualitative information. Thank you for this comment. 

We agree that in their current form, Tab. 2 and Tab. 3 are a little difficult to read. Due to the complex nature of the correlations which we present, however, it is not possible to present this information in the comprehensive form with comparative columns as we do in Tab. 1. Nonetheless, we have changed the format of Tab. 2 and 3, and created tables for each Germany (a) and South Africa (b). To make them more reader friendly, we emphasize relevant results in bold, and there are footnotes for additional explanations. 

It may also help if authors discuss the theoretical/practical implications of the research. Can future studies focus on the current study's limitations? How can the findings guide scientists who wish to become visible or are asked to speak in public about their research and/or findings? How do they maintain credibility given the public's perception of visible scientists? Thanks for pointing out this important point. We have added a paragraph on the limitations of our study on p.31, which also addresses options for future study. 

We have also emphasized throughout the paper the continuities in the role of visible scientists, as well as the large similarities between the two countries under study. As we show, now added on p. 27, credibility is first and foremost linked to academic reputation. At the same time, and as a practical advice to social media-active scientists, we have stressed the finding that the surveyed publics are not interested in details about the scientists’ private lives as a recommendation (p. 27/28).

---

## [Decision Letter · Decision Letter 1]

17 Nov 2024

PONE-D-24-24028R1“They never appear on TV and if they have, I might have missed that moment.” How publics in South Africa and Germany view visible scientistsPLOS ONE

Dear Dr. Rödder,

Thank you for submitting your manuscript to PLOS ONE. After careful consideration, we feel that it has merit but does not fully meet PLOS ONE’s publication criteria as it currently stands. Therefore, we invite you to submit a revised version of the manuscript that addresses the points raised during the review process. 

I invited a third reviewer with great expertise on the topic as I feel your manuscript could benefit from additional feedback. Although the decision is 'Minor Revision', I strongly encourage you to take their suggestions to heart and enrich the discussion of your findings in the light of other studies. Please also extend the limitation of your empirical design and specify which measures should be taken to avoid potential response bias, as suggested by Reviewer 3.

We look forward to receiving your revised manuscript.

Kind regards,

Adrian A. Diaz-Faes, PhD

Academic Editor

PLOS ONE

Journal Requirements:

Reviewers' comments:

Reviewer's Responses to Questions

**Comments to the Author**

1. If the authors have adequately addressed your comments raised in a previous round of review and you feel that this manuscript is now acceptable for publication, you may indicate that here to bypass the “Comments to the Author” section, enter your conflict of interest statement in the “Confidential to Editor” section, and submit your "Accept" recommendation.

Reviewer #2: All comments have been addressed

Reviewer #3: All comments have been addressed

2. Is the manuscript technically sound, and do the data support the conclusions?

Reviewer #2: Yes

Reviewer #3: Partly

3. Has the statistical analysis been performed appropriately and rigorously? 

Reviewer #2: Yes

Reviewer #3: Yes

4. Have the authors made all data underlying the findings in their manuscript fully available?

Reviewer #2: Yes

Reviewer #3: Yes

5. Is the manuscript presented in an intelligible fashion and written in standard English?

Reviewer #2: Yes

Reviewer #3: Yes

6. Review Comments to the Author

Reviewer #2: I appreciate the effort to address the comments. The revised manuscript is an improved version of the previous submission. The current paper presents concepts and findings in a clearer manner -- easier to understand and grasp. Authors were also better able to articulate limitations and implications.

Reviewer #3: The paper addresses a scientifically relevant topic (visible scientists) and offers an interesting analytical perspective. Its second version has undoubtedly improved the previous one; however, two main weaknesses persist. The first concerns the fact that by asking interviewees what they think about visible scientists without distinguishing between those who have long since passed away (for example, Einstein) and those still living (and thus active), a certain confusion is introduced that may distort the responses. Obviously, this issue cannot be resolved after the survey has already been conducted, but it should be addressed with greater awareness. The second weakness pertains to the connection between scientific reputation and media visibility. On one hand, the authors seem to ignore that it has been well documented how high visibility very often corresponds to low reputation (Neresini et al. 2023). On the other hand, it is obvious that people expect visible scientists to be competent and thus have a good reputation. Furthermore, the general public is rarely able to establish reputation according to the criteria of the scientific community; rather, they can infer it from media visibility (if a scientist appears in the media, then he/she must be competent). However, on this point, the conclusions reached by the authors remain almost exclusively descriptive, while a more in-depth discussion would be necessary.

7. PLOS authors have the option to publish the peer review history of their article (what does this mean?). If published, this will include your full peer review and any attached files.

Reviewer #2: No

Reviewer #3: No

---

## [Editor Report · Decision Letter 2]

20 Dec 2024

“They never appear on TV and if they have, I might have missed that moment.” How publics in South Africa and Germany view visible scientists

PONE-D-24-24028R2

Dear Dr. Rödder,

We’re pleased to inform you that your manuscript has been judged scientifically suitable for publication and will be formally accepted for publication once it meets all outstanding technical requirements.

Kind regards,

Adrian A. Diaz-Faes, PhD

Academic Editor

PLOS ONE
---

## [Editor Report · Acceptance letter]

26 Dec 2024

PONE-D-24-24028R2 

PLOS ONE

Dear Dr. Rödder, 

I'm pleased to inform you that your manuscript has been deemed suitable for publication in PLOS ONE. Congratulations! Your manuscript is now being handed over to our production team.

Kind regards, 

on behalf of

Dr. Adrian A. Diaz-Faes 

Academic Editor

PLOS ONE